# Maturation of the human striatal dopamine system revealed by PET and quantitative MRI

Bart Larsen 1✉, Valur Olafsson 2, Finnegan Calabro 1,3, Charles Laymon3,4, Brenden Tervo-Clemmens1, Elizabeth Campbell4, Davneet Minhas4, David Montez1, Julie Price5 & Beatriz Luna1

The development of the striatum dopamine (DA) system through human adolescence, a time of increased sensation seeking and vulnerability to the emergence of psychopathology, has been difficult to study due to pediatric restrictions on direct in vivo assessments of DA. Here, we applied neuroimaging in a longitudinal sample of n = 146 participants aged 12–30. R2′, an MR measure of tissue iron which co-localizes with DA vesicles and is necessary for DA synthesis, was assessed across the sample. In the 18–30 year-olds (n = 79) we also performed PET using [11C]dihydrotetrabenazine (DTBZ), a measure of presynaptic vesicular DA storage, and [11C]raclopride (RAC), an indicator of D2/D3 receptor availability. We observed decreases in D2/D3 receptor availability with age, while presynaptic vesicular DA storage (as measured by DTBZ), which was significantly associated with R2′ (standardized coefficient = 0.29, 95% CI = [0.11, 0.48]), was developmentally stable by age 18. Our results provide new evidence for maturational specialization of the striatal DA system through adolescence.

[1] Department of Psychiatry, University of Pittsburgh, Pittsburgh, PA 15213, USA. [2] NUBIC, Northeastern University, Boston, MA 02115, USA. [3] Department of Bioengineering, University of Pittsburgh, Pittsburgh, PA 15213, USA. [4] Department of Radiology, University of Pittsburgh, Pittsburgh, PA 15213, USA. [5] Massachusetts General Hospital, Harvard Medical School, Charlestown, MA 02129, USA. ✉email: bart.larsen@pennmedicine.upenn.edu

A dolescence is a unique stage of development between childhood and adulthood that is prominently characterized by a heightened reward drive and sensation seeking behavior, which while generally adaptive, can lead to risk-taking behaviors with potentially dangerous consequences (e.g. substance use, reckless behavior, unprotected sex)[1–3]. Adolescence also marks the emergence of major psychopathology associated with impairments in motivation and reward-driven behavior, including mood disorders, substance use disorders, and suicide, suggesting that the developmental processes guiding normative increases in reward-driven behavior may also confer vulnerability to mental illness[4,5]. These public health concerns have led to an impetus to understand the neurodevelopmental mechanisms underlying adolescent reward-driven behaviors[6].

Prominent neurodevelopmental models posit that adolescent behavior is driven by a developmental peak in dopamine (DA) processing in the striatal reward system, particularly in the nucleus accumbens (NAcc)[3,7,8]. The striatal DA system, however, is complex and multifaceted, with multiple factors influencing its function, including synthesis, vesicular transport, dopamine transporter (DAT), and multiple subtypes of DA receptors, each of which may have a distinct developmental trajectory. Rodent models have identified important developmental changes in several of these aspects of the DA system, including pubertal peaks in D1 and D2 receptor concentration, predominantly in the dorsal striatum[9–11], and developmental increases in DA concentration[12,13] and DAT levels[14,15] throughout puberty that peak or plateau later in adulthood. However, direct in vivo developmental studies in humans have been limited due to contraindications in pediatric populations for direct DA probes, such as positron emission tomography (PET), which is invasive as a small dose of a radiotracer is introduced intravenously. To address these limitations, here we use multimodal neuroimaging in a longitudinal sample (two visits, 18 months apart) in which we obtain an indirect measure of dopamine neurobiology with MRI for the entire sample (12–30 years of age) and two additional, concurrent PET imaging measures in the adult portion of the sample (18–30-year-old).

Our multimodal approach measured brain tissue iron concentration as a pediatric MRI compatible indirect measure of DA neurobiology. Iron is stored as ferritin in the brain[16] where it supports cellular respiration and myelination, as well as serving a critical role in the synthesis of monoamine neurotransmitters including DA[16,17]. Iron co-localizes with DA vesicles at the microscopic level and is a necessary co-factor for tyrosine hydroxylase, the rate-limiting step of DA synthesis[18,19]. As such, tissue iron has greatest concentration in the DA-rich basal ganglia and midbrain. Importantly, tissue iron is paramagnetic and can consequently be measured non-invasively using MRI. MRI studies of disorders that affect the mesolimbic and nigrostriatal DA systems, such as Parkinson's disease[19], Huntington's disease[17], ADHD[20], restless legs syndrome[21], and cocaine addiction[22] have identified tissue iron concentration as an indicator of DA neurobiology, including D2 receptor density, DA concentration, and DAT function. Supporting a link to DAergic maturation during adolescence, initial developmental studies have shown developmental changes in tissue iron through adolescence, including our own work[23,24]. A number of MR acquisitions have been created to quantitatively measure iron concentration in the brain, including T2 and T2* relaxation and quantitative susceptibility mapping[25,26]. One such measure is R2′, the reversible transverse relaxation rate $(R2' = 1/T2' = 1/T2* - 1/T2)$. R2′ is linearly related to postmortem tissue iron concentration[27] and has been used to quantify striatal tissue iron in patients with Parkinson's disease[28]. Similar measurements (e.g. R2*) have also been used to characterize tissue iron across the lifespan[29].

In the present study, we obtained R2′ across the entire sample, spanning from adolescence to adulthood (146 individuals age 12–30 years at visit one). In adults (18–30 years of age at visit one), we also obtained PET measures of striatal DA with radiotracers [11C]Dihydrotetrabenazine (DTBZ) and [11C]Raclopride (RAC), in the same session using a Siemens mMR scanner allowing us to assess age-related changes in DA from 18 to 30. Critically, we also assess the association between tissue iron and PET measures of DA. DTBZ binds to the vesicular monoamine transporter (VMAT2) and its binding potential (BP) is used to quantify presynaptic vesicular DA storage[30]. RAC binds to D2/D3 dopamine receptors (D2/3R) and its BP has been used as an indicator of D2/3R availability. As animal models of adolescent development suggest a peak in DA receptors during mid adolescence and a later peak or plateau in DA concentration, innervation, and synthesis during early adulthood in the caudate, putamen, and nucleus accumbens, we hypothesize developmental decreases in RAC binding and stability of DTBZ binding in our adult sample from 18–30 years of age. In support of our hypothesis, our results indicate that while vesicular DA is developmentally stable by age 18, striatal D2/3R availability decreases through young adulthood, particularly in the dorsal striatum. Importantly, we find that tissue iron concentration, which increases throughout adolescence before beginning to stabilize in adulthood, is positively associated with VMAT2, supporting its ability to provide an indirect measure of vesicular DA concentration. Together, these results provide novel in vivo human evidence reflecting specialization of the striatal DA system through adolescence.

## Results

**Development.** Linear mixed-effects models, covarying for sex, visit number, and motion during PET acquisitions (see Online Methods), showed unique age-related changes from 18–32 years of age across three striatal regions of interest (ROIs): the caudate, putamen, and NAcc (See Table 1). RAC BP significantly decreased from 18–32 years of age ($N = 78$, 129 sessions after QA) in the caudate ($\beta_{age^{-1}} = 0.42$, 95% CI = [0.22, 0.62], $p = 0.00004$, $p_{Bonferroni} = 0.00036$) and putamen ($\beta_{age^{-1}} = 0.29$, 95% CI = [0.09, 0.49], $p = 0.00518$, $p_{Bonferroni} = 0.047$) following an inverse (age$^{-1}$) function. The decrease was not significant in the NAcc ($\beta_{age^{-1}} = 0.21$, 95% CI = [0.00, 0.42], $p = 0.05$, $p_{Bonferroni} = 0.45$) (Table 1; Fig. 1). DTBZ BP was developmentally stable throughout 18–32 years of age ($N = 74$, 119 sessions after QA) in NAcc and putamen, and followed a quadratic function in the caudate ($\beta_{age} = -0.27$, 95% CI = [−0.51, −0.03], $p = 0.029$, $p_{Bonferroni} = 0.260$; $\beta_{age^2} = -0.24$, 95% CI = [0.09, 0.39], $p = 0.0026$, $p_{Bonferroni} = 0.0236$ (Table 1; Fig. 1). R2′, assessed across the entire sample ($N = 121$, 180 sessions after QA), significantly increased linearly with age in the putamen ($\beta_{age} = 0.74$, 95% CI = [0.63, 0.86], $p < 0.00001$, $p_{Bonferroni} < 0.00001$), and increased following an inverse age (age$^{-1}$) function in the caudate ($\beta_{age^{-1}} = -0.47$, 95% CI = [−0.62, −0.33], $p < 0.00001$, $p_{Bonferroni} < 0.00001$) and NAcc ($\beta_{age^{-1}} = -0.30$, 95% CI = [−0.47, −0.14], $p = 0.00033$, $p_{Bonferroni} = 0.00297$) (Table 1; Fig. 1). Age-related differences did not vary by sex for any measure.

**PET associations with R2′.** We used longitudinal mixed-effects models to test for an association between R2′ and each PET measure in the adult portion of the sample. We found that there was a significant positive between-subject association between R2′ and DTBZ BP ($N = 65$, number of sessions = 95 after QA, see online methods) in the NAcc ($\beta = 0.266$, 95% CI = [0.08, 0.45], $p = 0.005$, $p_{Bonferroni} = 0.030$; Table 2; Fig. 2a), but not the caudate

**Table 1 Linear mixed effects regression table for age models.**

| Measure | Region | Variable | Coefficient (95% CI) | $\beta$ (95% CI) | t | N | DF | p | $p_{Bonf}$ |
|---|---|---|---|---|---|---|---|---|---|
| Raclopride | Nucleus Accumbens | | | | | | | | |
| | | $Age^{-1}$ | 6.64 (−0.00, 13.28) | 0.21 (0.00, 0.42) | 1.98 | 126 | 121 | 0.05011 | 0.45099 |
| | | Sex | 0.01 (−0.08, 0.09) | 0.03 (−0.39, 0.44) | 0.12 | 126 | 121 | 0.90455 | 1.00000 |
| | | Visit | 0.04 (0.00, 0.08) | 0.20 (0.01, 0.39) | 2.12 | 126 | 121 | 0.03573 | 0.32157 |
| | | Motion | −0.33 (−0.57, −0.09) | −0.17 (−0.29 −0.05) | −2.76 | 126 | 121 | 0.00673 | 0.06057 |
| | Caudate | | | | | | | | |
| | | $Age^{-1}$ | **20.35 (10.85, 29.85)** | **0.42 (0.22, 0.62)** | **4.24** | **128** | **123** | **0.00004** | **0.00036** |
| | | Sex | −0.05 (−0.18, 0.07) | −0.17 (−0.56, 0.22) | −0.86 | 128 | 123 | 0.39312 | 1.00000 |
| | | Visit | 0.14 (0.07, 0.21) | **0.45 (0.21, 0.68)** | **3.77** | **128** | **123** | **0.00025** | **0.00225** |
| | | Motion | 0.43 (−0.01, 0.87) | 0.14 (0.00, 0.29) | 1.92 | 128 | 123 | 0.05660 | 0.50940 |
| | Putamen | | | | | | | | |
| | | $Age^{-1}$ | **12.94 (3.94, 21.94)** | **0.29 (0.09, 0.49)** | **2.85** | **128** | **123** | **0.00518** | **0.04662** |
| | | Sex | −0.07 (−0.19, 0.04) | −0.25 (−0.65, 0.15) | −1.25 | 128 | 123 | 0.21277 | 1.00000 |
| | | Visit | **0.09 (0.02, 0.17)** | **0.32 (0.06, 0.57)** | **2.46** | **128** | **123** | **0.01525** | 0.13725 |
| | | Motion | 0.12 (−0.31, 0.56) | 0.04 (−0.11, 0.20) | 0.56 | 128 | 123 | 0.57554 | 1.00000 |
| DTBZ | Nucleus Accumbens | | | | | | | | |
| | | $Age^{-1}$ | 2.99 (−3.61, 9.59) | 0.10 (−0.13, 0.34) | 0.90 | 118 | 113 | 0.37151 | 1.00000 |
| | | Sex | −0.07 (−0.16, 0.01) | −0.39 (−0.84, 0.06) | −1.74 | 118 | 113 | 0.08530 | 0.76770 |
| | | Visit | −0.003 (−0.05, 0.04) | −0.01 (−0.24,0.21) | −0.13 | 118 | 113 | 0.89877 | 1.00000 |
| | | Motion | −0.03 (−0.42, 0.35) | −0.01 (−0.18, 0.15) | −0.16 | 118 | 113 | 0.87160 | 1.00000 |
| | Caudate | | | | | | | | |
| | | Age | −0.23 (−0.36, −0.09) | −0.27 (−0.51, −0.03) | −2.21 | 119 | 112 | 0.02884 | 0.25959 |
| | | $Age^2$ | **0.004 (0.00, 0.01)** | **0.24 (0.09, 0.39)** | **3.08** | **119** | **112** | **0.00263** | **0.02364** |
| | | Sex | −0.08 (−0.20, 0.04) | −0.31 (−0.76, 0.13) | −1.39 | 119 | 112 | 0.16642 | 1.00000 |
| | | Visit | **0.07 (0.02, 0.12)** | **0.25 (0.08, 0.43)** | **2.85** | **119** | **112** | **0.00527** | **0.04745** |
| | | Motion | −0.10 (−0.55, 0.35) | −0.03 (−0.16, 0.10) | −0.44 | 119 | 112 | 0.65817 | 1.00000 |
| | Putamen | | | | | | | | |
| | | Age | 0.01 (−0.01 0.02) | 0.08 (−0.15, 0.32) | 0.72 | 117 | 111 | 0.47478 | 1.00000 |
| | | Sex | −0.06 (−0.17, 0.06) | −0.22 (−0.67, 0.23) | −0.96 | 117 | 111 | 0.33785 | 1.00000 |
| | | Visit | 0.05 (0.00, 0.10) | 0.20 (0.00, 0.40) | 2.01 | 117 | 111 | 0.04670 | 0.42030 |
| | | Motion | 0.33 (−0.13, 0.80) | 0.10 (−0.04, 0.25) | 1.42 | 117 | 111 | 0.15716 | 1.00000 |
| R2′ | Nucleus Accumbens | | | | | | | | |
| | | $Age^{-1}$ | **−45.48 (−69.99, −20.97)** | **−0.30 (−0.47, −0.14)** | **−3.66** | **178** | **174** | **0.00033** | **0.00297** |
| | | Sex | −0.29 (−0.92, 0.34) | −0.15 (−0.48, 0.18) | −0.90 | 178 | 174 | 0.36698 | 1.00000 |
| | | Visit | 0.37 (0.01, 0.74) | 0.19 (0.00, 0.38) | 2.01 | 178 | 174 | 0.04637 | 0.41733 |
| | Caudate | | | | | | | | |
| | | $Age^{-1}$ | **−41.14 (−53.79, −28.49)** | **−0.47 (−0.62, −0.33)** | **−6.42** | **178** | **174** | **<0.00001** | **<0.00001** |
| | | Sex | −0.44 (−0.76, −0.12) | −0.39 (−0.68, −0.10) | −2.68 | 178 | 174 | 0.00799 | 0.07191 |
| | | Visit | 0.08 (−0.14, 0.31) | 0.07 (−0.13, 0.28) | 0.72 | 178 | 174 | 0.47321 | 1.00000 |
| | Putamen | | | | | | | | |
| | | Age | **0.18 (0.16, 0.21)** | **0.74 (0.63, 0.86)** | **13.28** | **179** | **175** | **<0.00001** | **<0.00001** |
| | | Sex | −0.38 (−0.66, −0.11) | −0.30 (−0.51, −0.08) | −2.75 | 179 | 175 | 0.00656 | 0.05904 |
| | | Visit | 0.05 (−0.17, 0.26) | 0.04 (−0.13, 0.20) | 0.43 | 179 | 175 | 0.66650 | 1.00000 |

Differences in N and degrees of freedom reflect variations in sample size based on exclusions and outlier removal.
Bold font indicates the variable is significant after controlling for multiple comparisons (p < 0.05 Bonferroni corrected).
*DTBZ* [11C]Dihydrotetrabenazine $BP_{ND}$, *Raclopride* [11C]Raclopride $BP_{ND}$, $\beta$ standardized regression coefficient (beta), *N* number of sessions included.

or putamen (Table 2). The association between R2′ and RAC BP ($N = 68$, number of sessions = 98 after QA, see online methods) was not significant (Table 2). Considering the observed developmental effects present in many of these measures, we repeated these analyses after removing estimated age effects from each measure and testing whether individual differences with respect to age in R2′ and PET measures were associated (see Online Methods). We observed a similar pattern of results, such that there was a significant positive association between R2′ and DTBZ in the NAcc ($\beta = 0.28$, 95% CI = [0.09, 0.46], $p = 0.003$, $p_{Bonferroni} = 0.019$, linear mixed-effects model; Table 3; Fig. 2b). This relationship remained significant when adding sex, visit number, and estimated head motion as covariates in the model ($\beta = 0.29$, 95% CI = [0.11, 0.48], $p = 0.002$, $p_{Bonferroni} = 0.012$, linear mixed-effects model).

Having established a between-subject association between R2′ and DTBZ in the NAcc, we next tested whether longitudinal change in R2′ was associated with longitudinal change in DTBZ (i.e., a within-subject association) using a crossed lagged panel model. This is a critical analysis as a correlated within-subject change suggests a common mechanism of change in R2′ and DTBZ BP. This analysis was conducted on 30 participants (18–30 years of age at visit one; 60 total R2′ and DTBZ datasets) that passed our stringent quality criteria for NAcc R2′ *and* DTBZ BP data at both time points. Within the cross-lagged panel model (Fig. 2c), a significant correlation was observed between NAcc R2′ and NAcc DTBZ BP residualized change scores ($\beta = 0.441$, $z = 2.21$, $p = 0.027$; Fig. 2d), while controlling for autoregressive paths (NAcc R2′: $\beta = 0.716$, $z = 5.63$, $p < 0.01$; NAcc DTBZ: $\beta = 0.814$, $z = 7.92$, $p < 0.01$), suggesting a common underlying longitudinal mechanism of change. Correlated longitudinal change amongst NAcc R2′ and DTBZ was also observed when covarying age and head motion during the DTBZ acquisition ($\beta = 0.442$, $z = 2.10$, $p = 0.036$) as well as when we include all

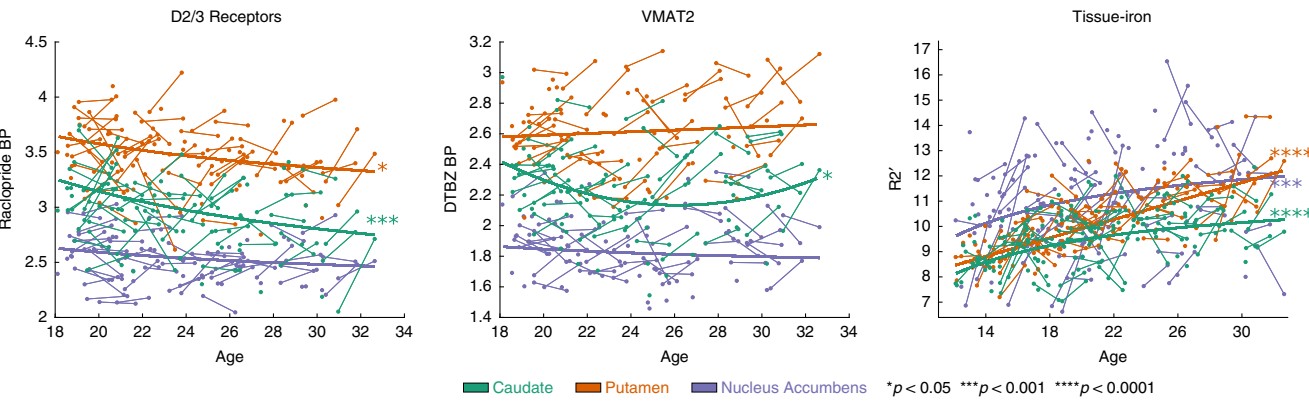

**Fig. 1 Developmental effects.** Age related-differences in Raclopride assessment of available D2/D3 receptor concentration ($N = 78$ individuals, 128 sessions; outlier removals: NAcc = 3, putamen = 1, caudate = 1), DTBZ assessment of VMAT2 concentration ($N = 74$ individuals, 119 sessions; outlier removals: NAcc = 1, putamen = 2, caudate = 0), and R2′ assessment of tissue iron content ($N = 121$ individuals, 180 sessions; outlier removals: NAcc = 2, putamen = 1, caudate = 2) in the striatum. All analyses were conducted using linear mixed-effects models. Model parameters can be found in Table 1.+$p <$ 0.05 uncorrected, *$p < 0.05$, ***$p < 0.001$, ****$p < 0.0001$ Bonferroni corrected. DTBZ BP [11C]Dihydrotetrabenazine binding potential; Raclopride BP [11C] Raclopride binding potential.

**Table 2 Linear mixed effects regression table for associations between R2′ and PET binding potentials.**

| PET measure | Region | R2′ coefficient (95% CI) | R2′ β (95% CI) | t | N | DF | p | $p_{Bonf}$ |
|---|---|---|---|---|---|---|---|---|
| Raclopride | Caudate | 0.029 (−0.034 0.091) | 0.090 (−0.11 0.29) | 0.90 | 98 | 96 | 0.371 | 1.000 |
| | Putamen | 0.004 (−0.050 0.057) | 0.013 (−0.19 0.22) | 0.13 | 98 | 96 | 0.898 | 1.000 |
| | Nucleus Accumbens | 0.017 (−0.005 0.040) | 0.162 (−0.03 0.36) | 1.66 | 98 | 96 | 0.100 | 0.600 |
| DTBZ | Caudate | 0.020 (−0.023 0.063) | 0.08 (−0.09 0.25) | 0.93 | 95 | 93 | 0.357 | 1.000 |
| | Putamen | 0.010 (−0.030 0.050) | 0.046 (−0.13 0.23) | 0.51 | 94 | 92 | 0.613 | 1.000 |
| | Nucleus Accumbens | **0.026 (0.008 0.040)** | **0.266 (0.08 0.45)** | **2.86** | **95** | **93** | **0.005** | **0.030** |

Differences in *N* and degrees of freedom reflect variations in sample size based on exclusions and outlier removal.
Bold font indicates the variable is significant after controlling for multiple comparisons ($p < 0.05$ Bonferroni corrected).
*DTBZ* [11C]Dihydrotetrabenazine BP$_{ND}$, *Raclopride* [11C]Raclopride BP$_{ND}$, β standardized regression coefficient (beta), *N* number of sessions included.

participants with available data for either measure at either time point and use full information maximum likelihood to impute missing data ($\beta = 0.434$, $z = 2.37$ $p = 0.018$; complete cases $\beta = 0.441$, $z = 2.21$, $p = 0.027$).

### Discussion
We used a multimodal neuroimaging approach to investigate developmental changes in indices of pre- and post-synaptic DA neurobiology through adolescence and young-adulthood. R2′ measures of tissue iron concentration showed increases throughout adolescence, replicating prior work[23,24,29,31,32] and were specifically associated with PET DTBZ measures of pre-synaptic vesicular dopamine. Given that DTBZ BP quantifies presynaptic vesicular DA storage[30], these findings suggest that DA availability, following the observed developmental pattern of R2′, increases through the adolescent period before stabilizing by early-adulthood. In contrast, PET RAC measures of post-synaptic DA receptor density showed decreases through adulthood.

These results are in agreement with and bring together findings from rodent models of pubertal development which indicate monotonic increases in DA levels and DAT levels during adolescence that begin to plateau in adulthood[12,14,15] and decreases in D1 and D2 receptor concentrations in the striatum through adulthood[9,11,15]. Specifically, our PET RAC findings indicate developmental decreases in available D2/3R concentration in the putamen and caudate in late adolescence and early adulthood (ages 18–32), supporting findings from animal models indicating age-related decreases in D2/D3 receptor concentration from adolescence to adulthood in these regions. Given that our PET RAC assessment is on 18–32 year-olds, this may reflect the

decline in receptors that follows a similar peri-adolescent peak observed in rodent models[9,11,15]. Further, similar to rodent models that find that NAcc DA receptor concentrations have either a less pronounced peak or a plateau during late adolescence[11,15] and recent findings from a human study of aging that showed a more shallow decrease in ventral striatum D2 receptors relative to caudate and putamen[33], we also found a less pronounced association with age in the NAcc for RAC BP. While this result is compelling and aligns well with prior literature, it is important to note that spatial resolution is a limitation of PET imaging, particularly in small volumes such as NAcc, and limited spatial resolution may influence the diminished age-related reductions observed in NAcc RAC BP. Though this is an important caveat, we took steps to address this issue including stringent quality control and motion correction procedures prior to analysis and the inclusion of in-scanner head motion as a covariate in statistical models. It is also important to note that though the RAC BP findings presented here replicate rodent models of a developmental reduction in D2/D3 receptor concentration, RAC BP is sensitive to *available* D2/D3 receptor concentration which can be affected by the level of endogenous DA binding. Thus, it is possible that developmental changes in endogenous DA binding can also impact available DA receptors and, as a result, impact RAC BP. Though it is not possible to fully disentangle these mechanisms with the available data, the relative developmental stability of DTBZ BP in the striatum during the same time suggests that changes in vesicular DA concentration are not driving the RAC BP effects. Future work is necessary to quantify the potential influence of developmental differences in *extracellular* DA concentration. Nevertheless, taken together, the

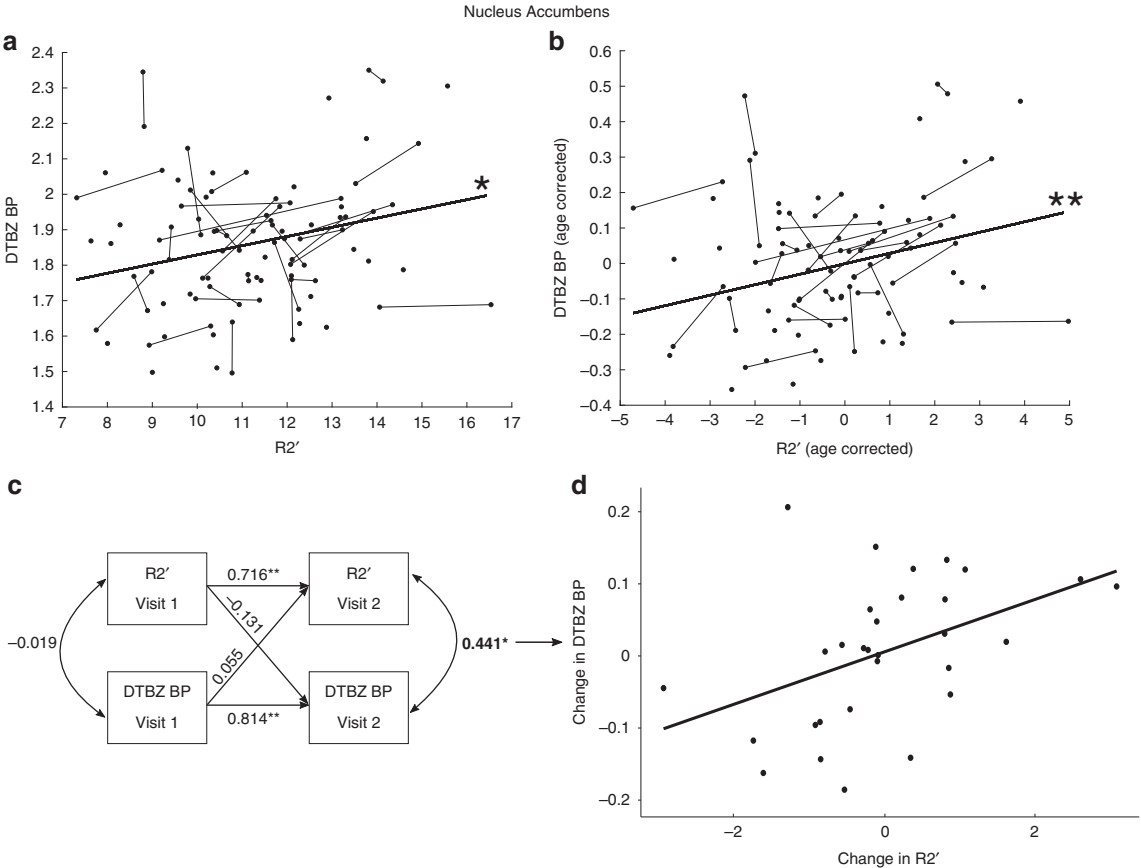

**Fig. 2 Association between R2′ and [11C]Dihydrotetrabenazine binding potential (DTBZ BP) in the nucleus accumbens. a** Tissue iron, assessed with R2′, is significantly positively associated with VMAT2 ($N = 98$ sessions), assessed with [11C]Dihydrotetrabenazine binding potential (DTBZ BP). Results are from a linear mixed-effects model (*$p < 0.05$ Bonferroni corrected). **b** This relationship remains after regressing age effects out of both variables (age corrected). **c** Cross-lagged structural equation model indicating within-subject residualized change in DTBZ BP and R2′ at visit two (bolded) are significantly positively correlated ($N = 30$ individuals, 60 sessions; *$p < 0.05$, **$p < 0.01$). **d** Visual depiction of the effect from **c**.

**Table 3 Linear regression table for associations between R2′ and PET binding potentials after residualizing with respect to age.**

| PET measure | Region | R2′(age corrected) coefficient (95% CI) | R2′(age corrected) $\beta$ (95% CI) | t | N | DF | p | $p_{Bonf}$ |
|---|---|---|---|---|---|---|---|---|
| Raclopride | Caudate | 0.046 (−0.016 0.011) | 0.15 (−0.05 0.35) | 1.47 | 98 | 96 | 0.146 | 0.876 |
| | Putamen | 0.039 (−0.025 0.102) | 0.12 (−0.08 0.31) | 1.21 | 98 | 96 | 0.231 | 1.000 |
| | Nucleus Accumbens | 0.020 (−0.002 0.042) | 0.18 (−0.02 0.37) | 1.82 | 98 | 96 | 0.071 | 0.427 |
| DTBZ | Caudate | 0.015 (−0.028 0.057) | 0.06 (−0.12 0.23) | 0.68 | 95 | 93 | 0.501 | 1.000 |
| | Putamen | −0.006 (−0.051 0.039) | −0.02 (−0.18 0.14) | −0.26 | 94 | 92 | 0.793 | 1.000 |
| | Nucleus Accumbens | **0.028 (0.010 0.046)** | **0.28 (0.09 0.46)** | **3.03** | **95** | **93** | **0.003** | **0.019** |

Differences in N and degrees of freedom reflect variations in sample size based on exclusions and outlier removal.
Bold font indicates the variable is significant after controlling for multiple comparisons ($p < 0.05$ Bonferroni corrected).
DTBZ [11C]Dihydrotetrabenazine BP$_{ND}$, Raclopride [11C]Raclopride BP$_{ND}$, $\beta$ Standardized regression coefficient (beta), N number of sessions included.

pattern of developmental results observed in this study provide in vivo evidence in humans for the development of pre- and post-synaptic DA processes that largely reflects earlier findings from rodent models (Fig. 3).

DTBZ BP provides an index of VMAT2, which packages and transports vesicular DA and is concentrated in presynaptic DA terminals in the striatum. As such, VMAT2 should be correlated with overall striatal DA concentration; however, it does not account for extracellular or cytosolic DA. The lack of developmental differences in DTBZ BP observed between ages 18 and 32 in this study may indicate that a developmental plateau in DA concentration has already been reached by late adolescence, a pattern supported by rodent models[12] (Fig. 3) as well as recent

work reporting developmental stability in DA synthesis capacity in the striatum during aging[34]. We did observe a significant quadratic effect in the caudate such that there was a small developmental trough during the mid-twenties, however, the linear age effect was not significant and suggests an overall pattern of stability that is consistent with findings from the putamen and NAcc. Though we cannot assess VMAT2 directly in the adolescent portion of our sample, the positive association between DTBZ BP and R2′ in the NAcc in the adult sample suggests the pronounced age-related increases in R2′ in NAcc during early adolescence may reflect, in part, increases in vesicular DA concentration.

Together, our results support a model of developmental specialization of the striatal DA system wherein maturation of

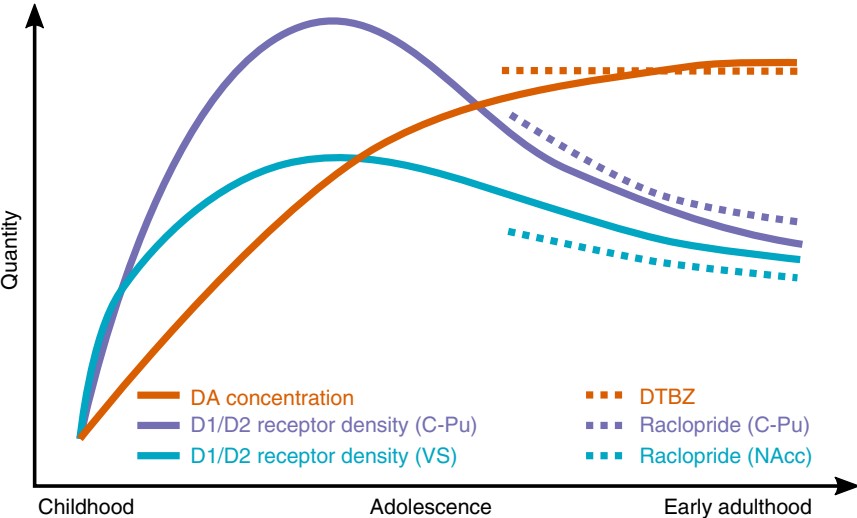

**Fig. 3 Schematic depiction of PET imaging results and prior work from rodent models of dopamine system development in the striatum.** Results from the present study are indicated in dashed lines, and schematic representations of age trajectories from developmental rodent models are depicted in solid lines. DA dopamine; DTBZ [11C]Dihydrotetrabenazine binding potential, Raclopride [11C]Raclopride binding potential; C-Pu caudate-putamen; VS ventral striatum; NAcc nucleus accumbens.

striatal DA availability precedes the more prolonged developmental reduction of available D2/3R concentration (Fig. 3), reflecting a process of refinement of DA circuitry that facilitates optimal, efficient DA signaling in adulthood. Importantly, this process is occurring at a time of known development of cognitive and reward-related functions[4,7]. Overall, the observed pattern of development reported here may reflect a period of plasticity that promotes specialization of striatal circuit neurobiology, similar to findings from studies of cortical circuit development that show synaptic pruning in prefrontal cortex during the same period[4,35]. An experience-dependent refinement in receptor concentration in the context of stable presynaptic DA concentration may promote more efficient and effective neural computation and thus enhanced performance for behaviors acquired during development—perhaps with some cost to the flexibility to form new associations and connections[11,36]. Considering that age-related reductions in RAC BP were diminished in the NAcc, this area may have neural and cognitive demands that necessitate a unique balance of DA concentration and receptor density that may favor persistent flexibility over specialization.

Intriguingly, the developmental processes observed here are occurring at the same time as other aspects of structural and functional development of the striatum. Human fMRI studies focusing on reward responses in the striatum have indicated that adolescents exhibit increased activation of putamen and NAcc in response to the receipt of rewards—and potentially to the anticipation of rewards—relative to children and adults[8]; however, it is important to note that some have shown reduced striatal responses[37] or no differences[38]. Cross-sectional[39,40] and longitudinal[41,42] studies of the development of striatal structural morphometry generally indicate decreasing striatal volume throughout adolescence and young-adulthood, particularly in the rostral-ventral striatum[40,42], and these ventral striatal volume changes are correlated with volume changes in other nodes of the motivation/affective circuit, including anterior cingulate cortex and orbitofrontal cortex[43]. At the same time, structural and functional connectivity between striatum and frontal brain areas involved in affective processing decrease during adolescence[44,45]. This prior work suggests that the developmental changes in the striatal DA system presented here are part of a larger developmental process whereby the structure and function of the

striatum is being dynamically sculpted into its stable adult configuration.

In addition to developmental findings, our approach provides evidence for tissue iron as a non-invasive, positive correlate of DA concentration. Our results demonstrated a significant positive relationship between tissue iron concentration, as measured using R2′, and VMAT2 content, measured with DTBZ in the NAcc. Critically, this relationship was observed as a between-subjects association, a within-subjects association (i.e. change in R2′ is associated with change in DTBZ), and as an individual differences association after removing the effect age from each measure, suggesting that tissue iron levels are associated with VMAT2 levels, an increase in tissue iron concentration is associated with an increase in VMAT2, and individual differences in tissue iron concentration with respect to age are associated with individual differences in VMAT2 with respect to age. The association between tissue iron and VMAT2 may reflect the necessary role of iron in DA synthesis[17,18], and supports a wealth of literature demonstrating correlative effects of iron concentration and DA neurobiology in disease[17,19–22]. These results may have important implications for researchers interested in assessing processes related to striatal DA in a non-invasive manner in populations unsuitable for PET imaging.

In contrast to NAcc, we did not observe significant associations between R2′ and DTBZ in the dorsal striatum. This is likely due to a combination of factors. One factor may be a more pronounced role for tissue iron in specifically supporting DA processes in NAcc. Two studies that investigated the effect of a rodent gene knockout model for iron regulation found that impaired iron regulation mechanisms led to reduced DAT, tyrosine hydroxylase, and VMAT2 in NAcc, but had minimal effects in dorsal striatum, indicating tissue iron levels may have particular significance for the regulation of DA neurobiology in NAcc[46,47]. Another likely contributing factor is the functional and neuroanatomical differences in the DA system between the dorsal and ventral striatum. Ventromedial striatum is involved in limbic processes and is predominantly innervated by VTA DA neurons whereas dorsolateral striatum is primarily involved in executive and sensorimotor processes and predominantly innervated by nigrostriatal pathways[48]. Ventromedial striatum also has less DAT uptake, less DA release, and a higher ratio of VMAT2/

uptake than caudal dorsolateral striatum[49–52]. Thus, dopaminergic processes in the NAcc have unique functional and neurobiological properties that may explain the pronounced relationship between tissue iron and DTBZ observed in this area.

Though these findings represent a significant advance in the understanding of the association between tissue iron concentration and striatal DA concentration, it is important to note limitations and opportunities for future directions. Though the positive associations we detect between R2′ and DTBZ, individual differences in R2′ and DTBZ with respect to age, and within-subject change in R2′ and DTBZ are compelling, it is important to note that we did not find a 1:1 relationship. As such, we do not rule out other sources of biological variability that can influence striatum iron concentration. Iron is critical for a number of essential biological processes, such as cellular respiration and myelination[16,17], which can impact striatal iron concentration. Future work investigating iron concentration in relation to measures of cerebral perfusion (such as arterial spin labeling) and myelin content in addition to DTBZ as well as animal models directly measuring DA and tissue iron would be beneficial in parsing the contributions of these additional sources of variability. Iron is also necessary for the synthesis of other monoamines, including serotonin and epinephrine, in addition to DA. Though these monoamines exist at much lower concentrations than dopamine in the striatum, we cannot address their relative contribution to the R2′ signal in this dataset. As such, future work is needed to replicate the current findings and assess the specificity of the between- and within-subject association between iron and DTBZ before R2′ can be confidently applied as a specific indicator of DA concentration. Accordingly, future studies using R2′ to investigate individual differences in striatal neurobiology must not narrowly interpret findings with respect to DA, and we caution against indicating it is a direct measure. Nevertheless, these results provide an encouraging advancement in this endeavor. Lastly, it is important to note that there was a high exclusion rate for R2′ scans based on our quality assessment criteria. Though exclusions were not age-dependent, the susceptibility of R2′ to macroscopic field inhomogeneity artifacts which can impact ventral aspects of the striatum (Supplementary Fig. 1) underscore the need for developing approaches to minimize artifacts and to quantify their impact.

In sum, this first-of-its-kind study provides novel in vivo human evidence for developmental specialization of critical DAergic processes, suggesting a model where DA availability is established through adolescence and may precipitate specialization of D2R receptor density. The complex changes in unique DAergic processes may underlie known developmental peaks in sensation seeking in adolescence. Importantly, the precise timing of these changes may also represent a period of vulnerability to impairment, possibly contributing to the emergence of psychopathology that is associated with the DAergic system (e.g., schizophrenia, mood disorders, addiction) that occurs at this time in development.

## Methods

**Sample**. One hundred forty-six adolescents and young adults participated in the study (ages 12–30; $M = 19.56$, SD = 5.01; 71 males, 75 females). Ninety-two participants also participated in a follow-up experimental session that occurred approximately 18 months ($M = 18.76$, $SD = 0.135$) after the first visit ($n = 146$ unique participants, 238 total sessions). The adult portion of the sample ($N = 79$; ages 18–30 years, $M = 23.29$, SD = 3.61; 39 males, 40 females) also participated in simultaneous PET imaging in addition to iron imaging (R2′) in order to assess indices of striatal DA neurobiology and compare them to tissue iron concentration. All participants reported no history of psychiatric or neurological disorder, and all participants provided informed consent. Informed consent was obtained from all participants. All experimental procedures in this study complied with the Code of Ethics of the World Medical Association (1964 Declaration of Helsinki) and were approved by the Institutional Review Board at the University of Pittsburgh. One

adult participant did not complete the DTBZ and R2′ acquisitions. In addition, following rigorous data quality assessment (described below), a total of 57 sessions were excluded from R2′ analyses, 21 sessions were excluded from DTBZ analyses, and 12 sessions were excluded from RAC analyses. Thus, we report on 47 12–17 year-olds and 78 18–30 year-olds.

**R2′ acquisition and estimation**. R2′ represents the reversible transverse relaxation rate (1/T2′) and is calculated as the difference between the effective (R2*; 1/T2*) and irreversible (R2; 1/T2) relaxation rates. This measure has been shown to be linearly related to tissue iron concentration[27]. In this study, we collected R2 using multi-echo turbo spin echo (mTSE) with the following parameters: effective echo times 12, 86, and 160 ms; TR = 6580 ms; 12 ms spacing between spin refocusing pulses; FoV = 240 × 240 mm$^2$; 27 3 mm transverse slices; 1 mm slice gap. R2* was calculated using multi-echo gradient echo (mGRE) with parameters: echo times 3, 8, 13, 18, and 23 ms; TR = 724 ms; flip angle, 25°; FoV = 240 × 240 mm$^2$. The R2′ image was estimated for each subject by estimating R2* and R2 images followed by a subtraction:

$$\widehat{R}_2' = \widehat{R}_2^* - \widehat{R}_2, \tag{1}$$

where the "hat" is used to indicate these are estimates. The acquired mGRE magnitude and phase images were used to estimate R2* and mTSE magnitude images to estimate R2. Since the mGRE echoes were acquired with alternating readout directions, thus causing distortion differences between the even and odd echoes, nonlinear registration was applied in the readout direction to undo the distortions.

Quadratic penalized least-squares (QPLS) was used to estimate either R2* or R2, similar to what was proposed in ref. [53], as follows:

$$\widehat{R} = \underset{R \in \mathbb{R}^N}{\arg\min} \left\{ \left( \sum_{j=1}^{n} \sum_{l=0}^{L} \frac{1}{2} \left| y_j^l - H_j^l \widehat{f}(R_j) e^{-R_j \Delta_l} \right|^2 \right) + \sum_{j=1}^{n} \frac{1}{2} \left| p(R_j) \right|^2 \right\}, \tag{2}$$

where $\widehat{R}$ is a vector with the estimated $R_2^*$ or $R_2$ pixel values, $j$ and $l$ are respectively spatial and echo difference indices, $y_j^l$ is either an mGRE or mTSE pixel value, $H_j^l$ is the through plane gradient pixel value as per[54], and similar to what was proposed in ref. [53] $\widehat{f}(R_j)$ is the maximum likelihood estimate of the spin density, $\Delta_l$ is the echo time difference, and $p(R_j)$ is the second order spatial difference of $R_j$ used for the penalty term to impose spatial smoothness on the estimate. When estimating R2* from mGRE data, $H_j^l$ was calculated from a field map estimate that used the mGRE phase images[53]. When estimating R2 from the mTSE data, through-plane gradients are minimal and were thus ignored. An iterative nonlinear conjugate gradient algorithm was used for the minimization[55] that was initialized by fitting the multi-echo data to a mono-exponential decay model[56].

Estimated R2* and R2 images were then registered to MNI space. This was performed using AFNI[57], where the R2 registration was performed by concatenating the affine registration between the first echo of the mTSE and anatomical image, and the non-linear registration of the anatomical image to MNI space. For the R2* registration a rigid-body registration between the first echoes of the mGRE and mTSE images was added to the concatenation. Finally, the R2′ image in MNI space was estimated by subtraction as discussed above.

**Image quality assessment**. All R2, R2*, and R2′ images were individually visually assessed by first author B.L. and co-author V.O. for data quality. The presence of motion, shimming, or registration artifacts were ranked on a five-point scale for severity. Visits with severity scores that exceeded three out of five, indicating that they produced a visually identifiable impact on the striatum, for any scan or had visible macroscopic field inhomogeneity artifacts in the R2′ image that affected the striatum were excluded from all analyses ($n = 57$). Notably, exclusions were not significantly dependent on participant age (parameter estimate = −0.34, CI = [−0.96, 0.26], $t = −1.11$, $p = 0.27$). See Supplementary Fig. 1 for examples of R2′ artifacts identified during quality assessment screening. The average of all included R2′ sessions is depicted in Supplementary Fig. 2.

**PET**. Tissue iron has been related to specific aspects of DA neurobiology, including D2 receptor density, dopamine transporter, synthesis, and energy production required for DA function. Our DTBZ and RAC PET indices allowed us to probe the relevance of tissue iron to two particular aspects of pre and postsynaptic DA processing. DTBZ binds to VMAT2 that transports monoamine neurotransmitters to presynaptic vesicles. DTBZ binding is a stable measure of presynaptic neuronal integrity in humans since over 95% of striatal VMAT2 binding sites are dopaminergic. In contrast, RAC binds to D2/D3 receptors throughout striatum, providing a postsynaptic index of available D2/D3 receptor concentration.

**Acquisition**. PET data were acquired on a Siemens mMR dual modality PET/MR scanner over two scanning sessions occurring in the same day. Participants completed a 90-min RAC acquisition followed by a 60-min DTBZ acquisition that occurred after a short break. Radioligands were administered with a Harvard syringe pump and a bolus + infusion (B + I) paradigm[58]. For RAC, participants

received a total injected dose of 33–40 mCi with high specific activity. The tracer was prepared in a 60 ml solution with 32.5 ml injected as the initial bolus over a period of 2 min. The remainder was injected at a constant rate of 18.57/h ml over the next 88 min (kbol = 105 min). For DTBZ, participants received a total injected dose of 15 mCi. The tracer was prepared in a 60 ml solution with 33.5 ml injected as the initial bolus over a period of 2 min with the remaining 26.5 ml injected at a constant rate over the next 58 min. Attenuation correction was performed using a combined segmentation- and atlas-based approach[59]. As part of the acquisition MR data were acquired and included a structural T1 Magnetization Prepared Rapid Gradient Echo (MPRAGE) sequence as well as an ultrashort TE (UTE) MR pulse sequence.

**Image generation**. We employed the template-based method and software of Izquierdo-Garcia et al[59]. to generate optimal[60] attenuation correction from the T1-MPRAGE image acquired simultaneously with the PET. The attenuation map was imported to the mMR and a PET reconstruction performed. The PET data, DTBZ data were reconstructed into 20 frames each of 180 s duration using Fourier Rebinning and Filtered Back Projection (FORE-FBP) (RAC: 30 × 180 s time frames; DTBZ: 20 × 180 s time frames) and smoothed with a 3 mm Hann filter. The final image grid size was 256 × 256 × 127(axial) with a voxel size of 1.22 × 1.22 × 2.03 mm³. PET data were corrected for random coincidences, attenuation, scatter, deadtime, and decay using the manufacturer's software.

**Post-processing**. The multiframe PET data were examined for interframe motion and, when needed, corrected using the motion correction functions of the software PMOD 3.6 (PMOD Technologies LLC., Zürich, Switzerland). Frames were then visually inspected to confirm alignment. Data were then aligned to each subject's MPRAGE, and the same linear and non-linear warp coefficients derived from the MPRAGE were applied to transform data to MNI space. For DTBZ data, The MPRAGE MR image was processed with FreeSurfer 5.3[61] with the goal of generating the regions of interest (ROIs) used in downstream modeling. FreeSurfer-generated regions were inspected for quality and manually adjusted if required.

**Kinetic modeling: Raclopride**. The 90-min RAC study included two parts: [1] an initial, pre-task portion while subjects were at rest, and [2] a later portion beginning 35–40 min into the scan during which subjects performed a reward task as part of a separate study. The current study focuses on the pre-task portion of the RAC acquisition. BP estimates were obtained voxelwise by fitting the entire time course and including a delta-BP (γ) term to account for changes in BP due to the task, based on previous models[62]. This was performed using a modified version of the simplified reference tissue model (SRTM) as implemented in MIAKAT[63]. Specifically, re-arrangement of the terms of Eq. 7 from Lammertsma and Hume[64] given $R_1 = k_1/k_2$ and $k_1/k_2 = k_1'/k_2'$ produces

$$C(t) = R_1 C'(t) + R_1 \left[ k_2' - \frac{R_1 k_2'}{1 + BP} \right] C'(t) \otimes e^{\left( \frac{-R_1 k_2' t}{1 + BP} \right)}. \quad (3)$$

In order to adapt this model to admit a time varying binding potential it may be re-expressed as an equivalent temporal update rule rather than a convolution, which produces

$$C(t) = R_1 C'(t) + R_1 \left[ k_2' - \frac{R_1 k_2'}{1 + BP} \right] W(t) \quad (4)$$

where

$$W(t) = e^{(-BP)} W(t-1) + \left[ \frac{1 - e^{(-BP)}}{BP} \right] C'(t). \quad (5)$$

This temporal update model is equivalent to the current MIAKAT implementation. Our modified implementation replaced BP with BP(t), modeled as a Heaviside step function in the following way.

$$BP(t) = BP_{ND} - \gamma h(t) \quad (6)$$

$$h(t) = \begin{cases} 0 (t \le t_{onset}) \\ 1 (t > t_{onset}) \end{cases} \quad (7)$$

In this expanded model, C(t) and C′(t) represent the time activity curves of the voxel of interest and reference region respectively, while $R_1$ is the ratio of tracer delivery ($k_1/k_1'$) and $k_2'$ is the reference region clearance rate constant. Modeling BP(t) as a Heaviside step function (Eqs. 5 and 6) allows for time-dependent binding potential having a constant pre-task value $BP_{ND}$, which decreases by a fixed quantity, γ, at the moment the task began ($t_{onset}$), and remains at that level for the remainder of the acquisition. Time activity curves (TAC) were extracted for each voxel in the striatum, as well as a regional average TAC derived from the cerebellar gray matter excluding the vermis which was used as the reference region. Least squares fits were performed using Matlab's lsqnonlin function to estimate four free parameters ($R_1$, $k_2'$, $BP_{ND}$, γ), of which the output variable of interest was the baseline (pre-task) $BP_{ND}$. Observed $BP_{ND}$ values were comparable to prior literature[65–70].

**Kinetic modeling: DTBZ**. The Pixelwise Modeling Tool of PMOD was used for performing kinetic modeling. Data were fitted using the Multilinear Reference Tissue Model 2 (MRTM2) of Ichise et al.[71]. The modeled activity concentration in tissue of interest, $C_{Mod}(t)$, as a function of time ($t$) is given by:

$$C_{Mod}(t) = -\frac{V_T}{V_T' b} \left( \int_0^t C_{ref}(t') dt' + \frac{1}{k_2'} C_{ref}(t) \right) + \frac{1}{b} \int_0^t C_T(t') dt' \quad (8)$$

where $C_{ref}$ and $C_T$ are tracer concentrations in reference tissue and in the tissue of interest respectively, $V_T'$ and $V_T$ are distribution volumes in the reference tissue and tissue of interest, $k_2'$ is the $k_2$-value of the reference tissue, and $b$ is a parameter. In this model $\frac{V_T}{V_T' b}$ and $\frac{1}{b}$ are the coefficients determined in a voxel-wise linear fit of the data. The reference region for the current work is based on a study by Chan et al.[72] who found the occipital cortex to be more reliable than the cerebellum for DTBZ. Chan et al. did not use an MR scan for anatomical reference, but instead applied a set of large circular ROIs directly to the PET. In the current work, in which ROIs were defined using FreeSurfer and an anatomical T1 MR, the reference region was constructed from the union of FreeSurfer gray and white matter regions ctx-lh-pericalcarine (region index 1021), ctx-rh-pericalcarine (2021), wm-lh-pericalcarine (3021), and wm-rh-pericalcarine (4021).

For each scan, a single value of $k_2'$ was used for all voxels and was determined from a *regional* MRTM analysis of the putamen, a region with high specific binding, constructed from FreeSurfer regions Left-Putamen (index 12) Right-Putamen (index 51), in which $k_2'$ was a fitted parameter. Estimated $k_2'$ values were similar across regions in the striatum (Putamen: M = 0.18, SD = 0.02; NAcc: M = 0.18, SD = 0.04; Caudate: M = 0.2, SD = 0.04). In this study, both voxel-wise and regional, the pericalcarine was used as reference tissue[72]. In general, Eq. (3) is valid only after an internal equilibrium time $t*$. The Pixelwise Modeling Tool has an option for identifying an appropriate value of $t*$ that was used in this work. The value of $t*$ was taken to be the smallest value of $t$ for which the maximum relative difference between the fit and any measured data point with $t \ge t*$ was less than 10%. For most scans, all data points ($n = 20$) were used in the fit of these bolus/infusion data. In a few cases the value $t*$ was such that the first data point was excluded and all others were used. The non-displaceable binding potential, i.e. binding potential relative to the pericalcarine reference region, at each voxel was calculated as $BP_{ND} = \frac{V_T}{V_T'} - 1$ where $\frac{V_T}{V_T'}$ is determined from the two fit coefficients described above. The resulting $BP_{ND}$ parametric images were used in subsequent analyses.

**Image quality assessment**. DTBZ and RAC data were examined for data quality. PET images with poor alignment to the standard MRI template or data that resulted in poor curve fits and binding potential estimates (model fit $R^2 < 0.75$) were excluded from future analyses (DTBZ $n = 21$ scans; RAC $n = 12$ scans) resulting in 119 DTBZ scans ($n = 74$ unique participants) and 129 RAC scans ($n = 78$ unique participants) used. See Supplementary Fig. 2 for average images of all included DTBZ and RAC sessions.

**Region of interest selection**. All analyses were performed in a priori striatal regions of interest (ROIs). Striatal ROIs included nucleus accumbens, putamen, and caudate and were identified using the Oxford-GSK-Imanova structural atlas[73]. These regions of interest were then applied to voxel-wise R2′ data and voxel-wise parametric PET data.

**Statistical approach**. We used linear mixed effects regression models (fitlme in MATLAB), modeling participant ID as a random effect, to assess the developmental trajectories of R2′ across the striatum. In order to detect potential non-linear effects of age, we first performed a model selection procedure in which we examined linear, quadratic, and age$^{-1}$ functional forms of age and selected the best-fitting model using BIC. The best-fitting functional form of age was used for subsequent modeling and significance testing. Each model included sex and visit number as covariates. We first tested for age × sex interactions, but as there were no significant interaction terms and interactions were removed from final models. Assessing motion during PET acquisitions is difficult; however, motion is highly rank-order correlated across subjects for within-session imaging acquisitions[74]. As such, to control for potential between-subject effects of motion in regression analyses that included PET data, we estimated motion on an fMRI scan that occurred during the corresponding PET acquisition and used it as a covariate. Extreme value outliers were detected as points with residuals with respect to age that exceeded three standard deviations from the mean and were not included in the final regression models. Standardized regression coefficients (β) were obtained by converting model variables to z scores based on the full sample mean and standard deviation. All significance tests were adjusted for nine comparisons (3 regions × 3 imaging acquisitions) using the Bonferroni correction.

To evaluate associations between R2′ and each PET measure, we again used linear mixed effects regression models with participant ID modeled as a random effect. To ensure that any multimodal associations were not being enhanced or suppressed by associations of each measure with age, we investigated the extent that individual differences (residualized with respect to age) in R2′ and PET measures were related. We regressed the best-fitting functional form of age and its

interaction out of both measures and tested for an association amongst the residuals. Finally, to ensure the robustness of the results, we repeated the analysis while also including sex, motion, and visit number as covariates. For all mixed effects models assessing the relationship between R2′ and each PET measure, extreme value outliers were detected as points that exceeded three standard deviations from the mean after removing age-effects for any neuroimaging measure and were not included in the final regression models. Standardized regression coefficients ($\beta$) were obtained by converting model variables to z scores based on the full sample mean and standard deviation. Each set of models were adjusted for six comparisons (3 regions × 2 PET datasets) using the Bonferroni correction.

Longitudinal associations between R2′ and dopamine measures were assessed examined using a cross-lagged panel model with maximum likelihood estimation (lavaan package, version 0.5–23.1097, in R, version 3.4.3). Following standard suggestions for model specification[75] the model included autoregressive and cross-lag coefficients and correlations between the measures at each visit. Primary analysis utilized complete cases (no missingness). Thirty participants (60 sessions) had full NAcc R2′ and DTBZ BP data at both time points after stringent quality assessment. Results were unchanged when utilizing full information maximum likelihood to include missing data.

**Reporting summary**. Further information on research design is available in the Nature Research Reporting Summary linked to this article.

## Data availability
The neuroimaging data that support the findings of this study are available in OpenNeuro with the identifier 10.18112/openneuro.ds002385.v1.0.1.

## Code availability
The code generated during the current study is available from the corresponding author on reasonable request.

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

## Acknowledgements
The project described was supported by Grant Number 5R01MH080243-07 from the National library of Medicine, National Institutes of Health.

## Author contributions
B.Larsen, B.Luna, and F. C. conceptualized the study; B.Larsen, B.Luna, F.C., C.L. and J.P. designed the study; B.Larsen, F.C., V.O., B.T-C., E.C., D.Minhas, and D.Montez analyzed the data; B.Larsen, V.O., C.L., F.C. and B.T-C. drafted the paper; B.Larsen J.P., and B.Luna revised the paper.

## Competing interests
The authors declare no competing interests.
