## [Peer Review File · Nature Communications]

Reviewers' Comments:

Reviewer #1:

Remarks to the Author:

General comments

The authors used multi-modal neuroimaging in a cohort of young healthy individuals. Imaging comprised [¹¹C]dihydrotrabenazine (DTBZ) positron-emission tomography (PET), [¹¹C]raclopride (RAC) PET and magnetic resonance imaging (MRI) of the reversible transverse relaxation rate (R2') as biomarkers of presynaptic vesicular dopamine (DA) storage, D2 and D3 DA receptor concentration and brain iron content, respectively. Central findings include a relatively stable vesicular DA and decreasing striatal D2/D3 concentrations with age in early adulthood as well as an association between R2' and DTBZ binding potential, which is assumed to be an index of the vesicular monoamine transporter 2 (VMAT2).

Overall, this is a very interesting combination of experiments. The results seem to support previous assumptions derived from animal research. Linking PET investigations of dopaminergic neurotransmission to R2' as a proxy for brain iron is of substantial current interest for potentially obtaining a non-invasive MRI-based biomarker of DA metabolism. Apart from this general enthusiasm, I see a number of specific issues (primarily methodological in addition to necessary clarifications), which are listed below and require careful consideration.

Specific points

1. While the authors refer to earlier work, in particular in animals, reporting age-related changes of different aspects of DA metabolism, their main hypotheses about particular changes and their directions in specified regions of interest (ROIs) is not sufficiently evident from the Introduction. For example, were the analyses limited to the nucleus accumbens (NAcc) and the dorsal striatum (caudate and putamen) to obtain a meaningful reduction of statistical tests or were other areas included in a more explorative fashion as well?
2. The authors should avoid suggesting interpretations of results that were insignificant after proper statistical tests. Considering the rather large number of comparisons, uncorrected p-values are not suitable indicators of true-positive findings (e.g., lines 103f: "[decreased] marginally in the NAcc ..." or lines 118f: "...there was a positive association between R2' and ..."). It is okay to mention uncorrected p-values in the tables (although the footnotes are ambiguous in whether corrected or uncorrected data are reported) as long as values after correction are also included to avoid potential misinterpretation.
3. In addition to the above comment, more details about the statistical approach should be added to the Online Methods. In particular, this relates to corrections for multiple comparisons. The Results section seems to suggest that a Bonferroni correction was employed, assuming that $n = 3$ null hypotheses are tested due to separate analyses of 3 ROIs. I note that the reported values for the caudate in line 101, $p = 0.00004$ uncorrected and $p = 0.0013$ corrected, do not seem to support this assumption; please clarify! Disregarding this potential discrepancy, it remains unclear why a factor of 3 would lead to a meaningful correction. The majority of tests aim at finding variations with respect to the factor age. Besides the 3 ROIs, analyses summarized in Table 1 further included 3 parameters (RAC, DTBZ and R2') and 3 'functional forms' of the factor age (linear, quadric and inverse), suggesting at least $n = 27$ null hypotheses that had been tested.
4. Apart from the above critique, numbers reported in Table 1 and in the Results section for the

identical analysis seem to differ: For age-related variations in R2', line 109 reports $p = 0.00518$ (uncorrected) and 0.0155 (corrected) for the caudate, whereas $p < 10^{-9}$ (uncorrected) is stated in Table 1. Apart from the fact that such remarkably large inconsistencies might challenge the reliability of the reported results, a value of $p = 0.00518$ as reported in the Results section would not "survive" a more rigorous multiple-comparison correction as suggested above with potential consequences for the interpretation.

5. As this is an imaging study, one would like to see, in general, typical examples of the obtained DTBZ and RAC PET images as well as R2' maps as demonstration of the achieved quality. These images might also be serve to indicate the selected ROIs.

6. Lines 20, 80, 95ff and 246ff: The information on the sample size and the number of sessions that is distributed over various parts of the manuscript (Abstract, Introduction, Results and Online Methods) is confusing and partly inconsistent: The Abstract and Online Methods suggest that a total of 146 subjects were included whereas the Introduction reports 149 subjects (probably a typo). The phrasing in the Abstract suggests a longitudinal multi-modal study (i.e. repeated measurements with both PET and MRI) in 146 subjects whereas the PET cohort comprised only 79 subjects (I acknowledge that this is correctly stated in the Abstract). Hence, only these measurements qualify as being 'multi-modal'. However, the number of subjects with repeated multi-modal sessions of acceptable quality was only 30. I admit, that this level of detail is not needed in the Abstract. Nevertheless, the Abstract should avoid "overselling" the sample size by only reporting the recruited population.

Finally, the Introduction suggests that both DTBZ and RAC acquisitions were performed in all PET sessions. For DTBZ, the authors report 118 sessions after QA plus 21 additional rejected sessions, that is, a total of 139 sessions. For RAC, there are 128 accepted and 12 rejected sessions, leading to an inconsistent number of 140 sessions. Please check.

7. Lines 296ff: While the definition of a rigorous image quality assessment (QA) is to be acknowledged, the procedures seem to be rather subjective. In particular, they are exclusively based on visual image inspection without a clear definition of the score, which limits reproducibility. It would be beneficial if the authors could suggest potential objective measures as indicators of insufficient quality (e.g., derived from signal-to-noise ratio or measures of artifact levels). One might also consider to demonstrate a few typical results of images of high quality (which were included in the analyses) in comparison to images of different degrees of insufficient quality (which were discarded) as supplementary information.

8. The number of rejected MRI results is relatively large. It is also not immediately clear, why the different modalities, which are all affected by motion and are acquired (in parts) simultaneously, yield quite different rejection rates (RAC: 9%, DTBZ: 15%, MRI: 32%). The authors should briefly comment on these observations. Would this indicate insufficient robustness in future applications of R2' as a biomarker? As PET scans were not performed in subjects aged below 18 years, it might be interesting to see if a higher rejection rate of MRI sessions resulted for the adolescents as compared to the adults.

9. Lines 12f: Affiliation number 5 (Massachusetts General Hospital) is not assigned to any of the authors. Either remove or it from the affiliations list or complete the author line accordingly.

10. I did not find (Tziorti et al. 2011) (line 409) and (Savalia et al. 2017) (line 418f) quoted in the Online Methods in the Reference list.

11. Please update reference 37, Tamnes et al., Cereb. Cortex 2010; 20: 534-548.

Reviewer #2:

Remarks to the Author:

This is a unique dataset and interesting approach whereby the authors attempt to combine data from multiple imaging measures (MRI and PET) in young adults to try to make inferences about dopamine based on the MRI data that spans adolescence and the early stages of young adulthood. The discovery of a MRI measure that could reliably index even any aspect of dopamine function would be revolutionary and would open the door to countless interesting studies in young people that can't be imaged with PET. The present findings are interesting but do not provide that evidence. Specific comments below.

There are a few terminology issues that need to be resolved. The most important one is that radioligand binding potential is not a measure of receptor concentration. The convention in PET is to use the phrase "receptor availability" since BPND can be high because receptor density is high or because DA concentration is high (creating competition between endogenous DA and ligand). This latter issue is critical and needs acknowledgement in the paper.

Related to terminology, what does "peak in dopamine (DA) processing" mean? Processing is an unclear term. Maybe something like function would be more general. What is an increase in DA "content"? Does that mean concentration or levels?

It is unclear whether the data support the claim of "evidence for an innovative non-invasive marker of DA concentration". This is a risky claim to make since some suggestive initial evidence will likely contribute to a proliferation of papers making specific claims about dopamine from MRI data alone. An example of this problem recently occurred with eye blink rate. The correlation between EBR and dopamine is weak and unreliable at best and non-existent at worst. The coefficients in the mixed effects models are not easy to evaluate. Standardized effect sizes with confidence intervals will need to be included to evaluate these inferences. At minimum a column with standardized effect sizes and CIs should be added to Tables 2 and 3. Any claims need to be qualified by the uncertainty of the estimates. This is especially important since most analyses in the paper have different sample sizes. There needs to be more clarity on why such a small subset of data were used to evaluate the PET and MRI correlations and why those specific covariates were used. The sample sizes should also be listed in all table and figure captions.

One of the justifications for the tissue iron concentration and DA link is anatomical since DA is so high in striatum. However, GABA concentration is much, much higher than DA in the striatum. In general, the strength of the potential reverse inference needs more clear consideration. Though there is some evidence that iron could be associated with some specific aspects of DA, similar associations could exist with between iron and different aspects of several other neuromodulators or neurotransmitters. The quite different developmental patterns for VMAT2 and tissue iron that are clear in Figure 1 raise questions about how these two measures could be indexing the same thing.

There is a suggestion that the dopamine system develops and reaches a plateau sometime in early adulthood. Rather, it seems that the only thing that may be stable across adulthood is synthesis capacity. Evidence from animals suggests that dopamine concentration declines with age even though synthesis may be preserved, so the increase in concentration into adulthood is also not quite right. Aside from synthesis, every other measure of DA changes from 18 to the end of life. See Karrer et al 2017 in *Neurobiology of Aging*. Figure 6 in that paper shows an aggregation of the subset of studies that have individual subject data and makes clear that the model of receptor plateau (depicted in the

model diagram at the end of your paper) probably doesn't happen. Related to aging, in general the PET effects presented here are consistent with widely published adult age differences. For example, the lack of an age effect in ventral striatal D2 receptor availability has been documented recently in Seaman et al 2019 Human Brain Mapping.

Given the lack of a plateau, I was hoping for more discussion or even analysis of potential non-linear effects. In fact, there may be enough early adulthood (18-30 yo) data from the Karrer et al meta-analysis to examine nonlinearity during that phase - maybe for both D2 receptors and transporters. Those data are all publicly shared and open for secondary analysis.

I didn't realize until the methods details that the rac scan was a task-displacement design. Individual differences in task ability can complicate the use of these measures as stable estimates of individual differences in receptor availability so it is good that the authors excluded the task part of the session. It's quite possible that 40 minutes of data approximates or maybe even almost exactly aligns with 60 minutes of data (which is common for raclopride). Do others have data showing this?

I want to make it clear that this is an incredibly innovative study. There are multiple PET measures which is rare and the longitudinal data is incredibly valuable. I, likely much more than most, would be extremely excited if this association existed. I recently moved from an institution with a full PET center to one without local radiochem. I am desperate for MR measures that would provide reliable markers of dopamine function. I worry about the claims currently made in this paper though. The information presented does not seem strong enough to warrant the claims.

Gregory Samanez-Larkin

Reviewer #1 (Remarks to the Author):

General comments

The authors used multi-modal neuroimaging in a cohort of young healthy individuals. Imaging comprised [11C]dihydrotetrabenazine (DTBZ) positron-emission tomography (PET), [11C]raclopride (RAC) PET and magnetic resonance imaging (MRI) of the reversible transverse relaxation rate (R2') as biomarkers of presynaptic vesicular dopamine (DA) storage, D2 and D3 DA receptor concentration and brain iron content, respectively. Central findings include a relatively stable vesicular DA and decreasing striatal D2/D3 concentrations with age in early adulthood as well as an association between R2' and DTBZ binding potential, which is assumed to be an index of the vesicular monoamine transporter 2 (VMAT2).

Overall, this is a very interesting combination of experiments. The results seem to support previous assumptions derived from animal research. Linking PET investigations of dopaminergic neurotransmission to R2' as a proxy for brain iron is of substantial current interest for potentially obtaining a non-invasive MRI-based biomarker of DA metabolism. Apart from this general enthusiasm, I see a number of specific issues (primarily methodological in addition to necessary clarifications), which are listed below and require careful consideration.

Specific points

1. While the authors refer to earlier work, in particular in animals, reporting age-related changes of different aspects of DA metabolism, their main hypotheses about particular changes and their directions in specified regions of interest (ROIs) is not sufficiently evident from the Introduction. For example, were the analyses limited to the nucleus accumbens (NAcc) and the dorsal striatum (caudate and putamen) to obtain a meaningful reduction of statistical tests or were other areas included in a more explorative fashion as well?

We appreciate the opportunity expand on and clarify our hypothesis and rationale. We now clarify that our focus is specifically on the striatum for two reasons. First, we sought to test a hypothesis of theoretical models of human dopaminergic development of the striatum that suggests a peak in dopamine receptor concentration during adolescence and a developmental plateau of dopamine content during early adulthood. This hypothesis is based on a wealth of literature from rodent studies that we schematize in figure 3. Given that our PET dataset is limited to the adult portion of the sample, we hypothesize to find a developmental reduction in RAC binding (reflecting receptor availability) and stability of DTBZ binding (reflecting vesicular DA content). We now more explicitly emphasize these hypotheses about the direction of our PET effects in the Introduction. The second reason we restricted analyses to the striatum is that our PET measures have high signal-to-noise ratio in the striatum and low signal-to-noise outside of the striatum. This is illustrated in the addition of Supplementary Figure 2, which we have added based on the reviewer's suggestion (R2 #5).

“As animal models of adolescent development suggest a peak in DA receptors during mid adolescence and a developmental plateau in DA concentration during early adulthood in the caudate, putamen, and nucleus accumbens, we hypothesize developmental decreases in RAC binding and stability of DTBZ binding in our adult sample from 18-30 years of age.”

Supplementary Figure 2. Average images for all included R2', [11C]Raclopride BP_{ND} (RAC), and [11C]Dihydrotrabenazine BP_{ND} (DTBZ) sessions. Blue outline indicates the striatum.

2. The authors should avoid suggesting interpretations of results that were insignificant after proper statistical tests. Considering the rather large number of comparisons, uncorrected p-values are not suitable indicators of true-positive findings (e.g., lines 103f: “[decreased] marginally in the NAcc ...” or lines 118f: “...there was a positive association between R2' and ...”). It is okay to mention uncorrected p-values in the tables (although the footnotes are ambiguous in whether corrected or uncorrected data are reported) as long as values after correction are also included to avoid potential misinterpretation.

We appreciate these comments and have edited the text (below) in the lines identified to emphasize that these effects are not statistically significant and do not interpret them in the discussion section.

“The decrease was not significant in the NAcc.”

“The association between R2' and RAC BP... was not significant.”

We have also clarified the p-value reporting in all tables to include uncorrected and Bonferroni corrected p-values. As mentioned below, we also added standardized coefficients and confidence intervals to the tables to clarify the reporting of all results.

3. In addition to the above comment, more details about the statistical approach should be added to the Online Methods. In particular, this relates to corrections for multiple comparisons. The Results section seems to suggest that a Bonferroni correction was employed, assuming that $n = 3$ null hypotheses are tested due to separate analyses of 3 ROIs. I note that the reported values for the caudate in line 101, $p = 0.00004$ uncorrected and $p = 0.0013$ corrected, do not seem to support this assumption; please clarify! Disregarding this potential discrepancy, it remains unclear why a factor of 3 would lead to a meaningful correction. The majority of tests aim at finding variations with respect to the factor age. Besides the 3 ROIs, analyses summarized in Table 1 further included 3 parameters (RAC, DTBZ and R2') and 3 ‘functional

forms' of the factor age (linear, quadric and inverse), suggesting at least $n = 27$ null hypotheses that had been tested.

In line with Reviewer comments, we have readjusted our multiple comparison criteria in the Results section and our rationale in the Method section. We agree that a more precise approach is 3 (ROI) x 3 (imaging acquisitions) = 9 comparisons for our developmental models ($p < 0.0056$). We have now also adjusted for 3 (ROI) x 2 (PET modalities) = 6 comparisons in our multimodal models ($p < 0.008$). We now describe this in our Method section and reported both uncorrected and corrected p-values for all in-text results and all tables.

“All significance tests were adjusted for nine comparisons (3 regions x 3 imaging acquisitions) using the Bonferroni correction.”

“[PET associations with R2'] models were adjusted for six comparisons (3 regions x 2 PET datasets) using the Bonferroni correction.”

We also wish to clarify that the 3 functional forms of age were only examined with respect to model selection. No significance testing was performed at this stage and thus we do not perform correction for this model selection procedure, which is widely used in the developmental literature. We have clarified this in the method section as follows:

“We first performed a model selection procedure in which we examined linear, quadratic, and age⁻¹ functional forms of age and selected the best-fitting model using BIC. The best fitting functional form of age was used for subsequent modeling and significance testing.”

4. Apart from the above critique, numbers reported in Table 1 and in the Results section for the identical analysis seem to differ: For age-related variations in R2', line 109 reports $p = 0.00518$ (uncorrected) and 0.0155 (corrected) for the caudate, whereas $p < 10^{-9}$ (uncorrected) is stated in Table 1. Apart from the fact that such remarkably large inconsistencies might challenge the reliability of the reported results, a value of $p = 0.00518$ as reported in the Results section would not “survive” a more rigorous multiple-comparison correction as suggested above with potential consequences for the interpretation.

In light of these comments, we have now thoroughly checked all reported values for accuracy. We have corrected the typo in the reporting of age effects in the results section (and curtailed the number of significant digits reported). It now corresponds to Table 1 and reads:

“... putamen ($\beta_{\text{age}^{-1}} = 0.29$, 95% CI = [0.09, 0.49], $p = .00518$, $p_{\text{Bonferroni}} = .047$)...”

“...caudate ($\beta_{\text{age}^{-1}} = -0.47$, 95% CI = [-0.62, -0.33], $p < .00001$, $p_{\text{Bonferroni}} < .00001$)...”

As such, this effect survives the multiple comparison correction described above.

5. As this is an imaging study, one would like to see, in general, typical examples of the obtained DTBZ and RAC PET images as well as R2' maps as demonstration of the achieved quality. These images might also be serve to indicate the selected ROIs.

We agree that this would be helpful and now display the average images for each modality across all included sessions in Supplementary Figure 2 (see above).

6. Lines 20, 80, 95ff and 246ff: The information on the sample size and the number of sessions that is distributed over various parts of the manuscript (Abstract, Introduction, Results and Online Methods) is confusing and partly inconsistent: The Abstract and Online Methods suggest that a total of 146 subjects were included whereas the Introduction reports 149 subjects (probably a typo).

The phrasing in the Abstract suggests a longitudinal multi-modal study (i.e. repeated measurements with both PET and MRI) in 146 subjects whereas the PET cohort comprised only 79 subjects (I acknowledge that this is correctly stated in the Abstract). Hence, only these measurements qualify as being 'multi-modal'. However, the number of subjects with repeated multi-modal sessions of acceptable quality was only 30. I admit, that this level of detail is not needed in the Abstract. Nevertheless, the Abstract should avoid "overselling" the sample size by only reporting the recruited population.

We now better clarify in the introduction and Abstract that the total neuroimaging sample was 146.:

We applied neuroimaging in a longitudinal sample (146 12-30 year-olds) to assess developmental changes in DAergic processes

Finally, the Introduction suggests that both DTBZ and RAC acquisitions were performed in all PET sessions. For DTBZ, the authors report 118 sessions after QA plus 21 additional rejected sessions, that is, a total of 139 sessions. For RAC, there are 128 accepted and 12 rejected sessions, leading to an inconsistent number of 140 sessions. Please check.

We now clarify that one participant did not complete the R2' and DTBZ portions of the study.

"One adult participant did not complete the DTBZ and R2' acquisitions."

7. Lines 296ff: While the definition of a rigorous image quality assessment (QA) is to be acknowledged, the procedures seem to be rather subjective. In particular, they are exclusively based on visual image inspection without a clear definition of the score, which limits reproducibility. It would be beneficial if the authors could suggest potential objective measures as indicators of insufficient quality (e.g., derived from signal-to-noise ratio or measures of artifact levels). One might also consider to demonstrate a few typical results of images of high quality (which were included in the analyses) in comparison to images of different degrees of insufficient quality (which were discarded) as supplementary information.

Given the unavailability of established quantitative QA methods for R2' such as are available in fMRI, we applied this rigorous yet subjective QA approach based on known MRI artifacts including motion, shimming, and macroscopic field inhomogeneities which produce visually identifiable markers, which we now detail in the Methods including examples in Supplementary Figure 1. We now elaborate that the scoring system we developed assessed the visual impact of artifact on the striatum and that we excluded all scans where artifact compromised the striatum. This conservative approach resulted in greater exclusions compared to PET measures due to the susceptibility of R2' to macroscopic field inhomogeneity artifacts (Supplementary Figure 1C). Our rigorous QA procedure resulted in images predominantly free of artifacts as shown in Supplementary Figure 2 which depicts the average image of all scans that passed QA.

Supplementary Figure 1. Examples of artifacts present in R2' data that were identified during quality assessment. All R2' scans were assessed for data quality. Scans that contained artifacts related to (A) motion, (B) shimming, or (C) macroscopic field inhomogeneity that impacted the striatum were excluded from all analyses. The outline of the striatum is indicated in blue. The red arrows indicate examples of areas where artifacts impact striatum.

8. The number of rejected MRI results is relatively large. It is also not immediately clear, why the different modalities, which are all affected by motion and are acquired (in parts) simultaneously, yield quite different rejection rates (RAC: 9%, DTBZ: 15%, MRI: 32%). The authors should briefly comment on these observations. Would this indicate insufficient robustness in future applications of R2' as a biomarker? As PET scans were not performed in subjects aged below 18 years, it might be interesting to see if a higher rejection rate of MRI sessions resulted for the adolescents as compared to the adults.

The relatively high rate of exclusion for R2' is a result of its susceptibility to other sources of artifact beyond motion. These additional artifacts are now depicted in Supplementary Figure 1 (above). Importantly, the rejection rate is not associated with age in this sample and we now report this in the text:

“Notably, exclusions were not significantly dependent on participant age (*parameter estimate* = -0.34, *CI* = [-0.96, 0.26], *t* = -1.11, *p* = .27).”

Further, we now address this as a limitation in possible future applications of R2' in our expanded Discussion section:

“Lastly, it is important to note that there was a high exclusion rate for R2' scans based on our quality assessment criteria. Though exclusions were not age-dependent, the susceptibility of R2' to macroscopic field inhomogeneity artifacts which can impact ventral aspects of the striatum (Supplementary Figure 1) underscore the need for developing approaches to minimize artifacts and to quantify their impact.

9. Lines 12f: Affiliation number 5 (Massachusetts General Hospital) is not assigned to any of the authors. Either remove or it from the affiliations list or complete the author line accordingly.

Thank you. This has been fixed.

10. I did not find (Tziorti et al. 2011) (line 409) and (Savalia et al. 2017) (line 418f) quoted in the Online Methods in the Reference list.

Thank you. These citations have been added to the reference list.

11. Please update reference 37, Tamnes et al., *Cereb. Cortex* 2010; 20: 534-548.

This has been updated in the reference list.

Reviewer #2 (Remarks to the Author):

This is a unique dataset and interesting approach whereby the authors attempt to combine data from multiple imaging measures (MRI and PET) in young adults to try to make inferences about dopamine based on the MRI data that spans adolescence and the early stages of young adulthood. The discovery of a MRI measure that could reliably index even any aspect of dopamine function would be revolutionary and would open the door to countless interesting studies in young people that can't be imaged with PET. The present findings are interesting but do not provide that evidence. Specific comments below.

We appreciate the reviewer's comments on the novelty of our study and greatly welcome identifying the important limitations and considerations we need to integrate when presenting these novel findings. As detailed in the responses below, we now delineate our rationale for underscoring the potential for R2' tissue iron to inform aspects of DA neurobiology in the context of development--an important advancement in identifying neurobiological mechanisms that underlie brain maturation. We emphasize that while we do not have evidence that R2' tissue iron is a direct measure of DA, as tissue iron also underlies myelination and the production of other monoamines, its predominance in the striatum and its association with DTBZ indicates a particularly close association and provides a novel manner to form and test new hypotheses. As these findings will have a large impact on the field, we also emphasize that future researchers exercise caution and avoid narrowly interpreting findings regarding R2'. We also suggest that animal models that can investigate the link between DA neurobiology and R2' tissue iron more directly are critically needed to further characterize this association, which the present study is not able to do. We have made substantial edits and additions to the text to address these points, which we delineate below.

There are a few terminology issues that need to be resolved. The most important one is that radioligand binding potential is not a measure of receptor concentration. The convention in PET is to use the phrase "receptor availability" since BPND can be high because receptor density is high or because DA concentration is high (creating competition between endogenous DA and ligand). This latter issue is critical and needs acknowledgement in the paper.

We thank the reviewer for highlighting this important distinction and have now changed all mentions of PET assessments with Raclopride of "receptor concentration" to either "receptor availability" or "available receptor concentration". We also acknowledge the potential impact of developmental changes in DA concentration on the development of Raclopride BPND:

“It is also important to note that though the RAC BP findings presented here replicate rodent models of a developmental reduction in D2/D3 receptor concentration, RAC BP is sensitive to *available* D2/D3 receptor concentration which can be affected by the level of endogenous DA binding. Thus, it is possible that developmental changes in endogenous DA binding can also impact available DA receptors and, as a result, impact RAC BP. Though it is not possible to fully disentangle these mechanisms with the available data, the relative developmental stability of DTBZ BP in the striatum during the same time suggests that changes in vesicular DA concentration are not driving the RAC BP effects. Future work is necessary to quantify the potential influence of developmental differences in *extracellular* DA concentration on RAC BP.”

Related to terminology, what does "peak in dopamine (DA) processing" mean? Processing is an unclear term. Maybe something like function would be more general. What is an increase in DA "content"? Does that mean concentration or levels?

We have now sharpened our terminology according to these helpful suggestions. We have replaced “DA processing” with “DA function” and “DA content” with “DA concentration”.

It is unclear whether the data support the claim of "evidence for an innovative non-invasive marker of DA concentration". This is a risky claim to make since some suggestive initial evidence will likely contribute to a proliferation of papers making specific claims about dopamine from MRI data alone. An example of this problem recently occurred with eye blink rate. The correlation between EBR and dopamine is weak and unreliable at best and non-existent at worst. The coefficients in the mixed effects models are not easy to evaluate. Standardized effect sizes with confidence intervals will need to be included to evaluate these inferences. At minimum a column with standardized effect sizes and CIs should be added to Tables 2 and 3. Any claims need to be qualified by the uncertainty of the estimates. This is especially important since most analyses in the paper have different sample sizes. There needs to be more clarity on why such a small subset of data were used to evaluate the PET and MRI correlations and why those specific covariates were used. The sample sizes should also be listed in all table and figure captions.

We agree that the novel findings presented here will have great impact on the field, and as a result it is important to cautiously and responsibly interpret them to not encourage weak claims of tissue iron being a direct measure of dopamine. First, to increase the clarity and interpretability of all reported effects, we have now added standardized effects and confidence intervals to all tables and in-text reporting of statistical models. We now also include sample sizes with all tables and figures.

For PET and MRI associations, we emphasize that the covariates were included to confirm the robustness of the results (i.e. that they were not explained by spurious motion or visit effects):

“Finally, to ensure the robustness of the results, we repeated the analysis while also including sex, motion, and visit number as covariates.”

We also report the effects with and without the addition of covariates:

“We observed a similar pattern of results, such that there was a significant positive association between R2’ and DTBZ in the NAcc ($\beta = 0.28$, 95% CI = [0.09, 0.46], $p = .003$, $p_{Bonferroni} = .019$;

Table 3; Figure 2B). This relationship remained significant when adding sex, visit number, and estimated head motion as covariates in the model ($\beta = 0.29$, 95% CI = [0.11, 0.48], $p = .002$, $p_{\text{Bonferroni}} = .012$).

We now also emphasize that our within-subject analyses were restricted to 30 adults (60 sessions) based on the requirement that all individuals pass our stringent quality assessment criteria for both R2' and DTBZ at both baseline and follow-up. This was done to minimize the influence of poor data quality obscuring true effects or leading to false positives. As can be seen with the addition of supplementary figure 1 (above), a limitation of R2' is macroscopic field inhomogeneity artifacts that can impact signal in the ventral striatum, and we felt that stringent quality assessment of R2' data was necessary to ensure confidence in the reported effects. Notably, the results of this analysis are unchanged when we include all subjects with available data for either measure at either time point and use full information maximum likelihood to impute missing data. Specifically, the result of correlated residualized change among DTBZ and R2' allowing for missing data were highly similar to the main result presented in Figure 2D (missingness allowed $\beta = .434$, $z = 2.37$ $p = .018$; complete cases $\beta = .441$, $z = 2.21$, $p = .027$). We now report this directly in the Results:

“[Correlated longitudinal change amongst NAcc R2' and DTBZ was also observed] when we include all participants with available data for either measure at either time point and use full information maximum likelihood to impute missing data ($\beta = .434$, $z = 2.37$ $p = .018$; complete cases $\beta = .441$, $z = 2.21$, $p = .027$).

Lastly, we have attenuated the language used to describe the effects observed here, e.g.: “In addition to developmental findings, our approach provides evidence for tissue iron as a non-invasive, positive correlate of DA concentration”,

and, as described in greater detail in the response below, we have expanded our discussion of the limitations of the current findings to encourage caution to future researchers.

“Though the positive associations we detect between R2' and DTBZ, individual differences in R2' and DTBZ with respect to age, and within-subject change in R2' and DTBZ are compelling it is important to note that we did not find a 1:1 relationship. As such, we do not rule out other sources of biological variability that can influence striatum iron concentration. ... Accordingly, future studies using R2' to investigate individual differences in striatal neurobiology must not narrowly interpret findings with respect to DA, and we caution against indicating it is a direct measure.”

One of the justifications for the tissue iron concentration and DA link is anatomical since DA is so high in striatum. However, GABA concentration is much, much higher than DA in the striatum. In general, the strength of the potential reverse inference needs more clear consideration. Though there is some evidence that iron could be associated with some specific aspects of DA, similar associations could exist with between iron and different aspects of several other neuromodulators or neurotransmitters. The quite different developmental patterns for VMAT2 and tissue iron that are clear in Figure 1 raise questions about how these two measures could be indexing the same thing.

This point is well taken and we appreciate the need to expand our discussion on this subject. While there is a wealth of literature from studies of Restless Legs Syndrome, Parkinson's, Huntington's, addiction, ADHD, and iron deficiency demonstrating links (which range from

correlative to approaching causal) between striatum (and midbrain) iron concentration to multiple aspects of dopamine as well as a known requirement of iron for dopamine synthesis, it is true that iron is vital to a number of other neurobiological processes. As the reviewer mentions, the developmental trajectories of DTBZ and R2' are not identical, possibly reflecting these additional sources of variability. We note, however, that the relationship between R2' and DTBZ is strengthened after removing the modeled age effects from each measure, suggesting a greater association between individual differences with respect to age (e.g. Table 3). As such, we agree it is likely that other factors can influence individual differences in iron concentration, and that future studies using R2' must exercise caution and consider these factors when interpreting data. We have added substantial text to the Discussion section to address these issues:

“Though these findings represent a significant advance in the understanding of the association between tissue iron concentration and striatal DA concentration, it is important to note limitations and opportunities for future directions. Though the positive associations we detect between R2' and DTBZ, individual differences in R2' and DTBZ with respect to age, and within-subject change in R2' and DTBZ are compelling it is important to note that we did not find a 1:1 relationship. As such, we do not rule out other sources of biological variability that can influence striatum iron concentration. Iron is critical for a number of essential biological processes, such as cellular respiration and myelination (Connor and Menzies, 1996; Ward et al., 2014) which can impact striatal iron concentration. Future work investigating iron concentration in relation to measures of cerebral perfusion (such as arterial spin labeling) and myelin content in addition to DTBZ as well as animal models directly measuring DA and tissue iron, would be beneficial in parsing the contributions of these additional sources of variability. Iron is also necessary for the synthesis of other monoamines, including serotonin and epinephrine, in addition to DA. Though these monoamines exist at much lower concentrations than dopamine in the striatum, we cannot address their relative contribution to the R2' signal in this dataset. As such, future work is needed to replicate the current findings and assess the specificity of the between- and within-subject association between iron and DTBZ before R2' can be confidently applied as a specific indicator of DA concentration. Accordingly, future studies using R2' to investigate individual differences in striatal neurobiology must not narrowly interpret findings with respect to DA, and we caution against indicating it is a direct measure. Nevertheless, these results provide an encouraging advancement in this endeavor. Lastly, it is important to note that there was a high exclusion rate for R2' scans based on our quality assessment criteria. Though exclusions were not age-dependent, the susceptibility of R2' to macroscopic field inhomogeneity artifacts which can impact ventral aspects of the striatum (Supplementary Figure 1) underscore the need for developing approaches to minimize artifacts and to quantify their impact.”

There is a suggestion that the dopamine system develops and reaches a plateau sometime in early adulthood. Rather, it seems that the only thing that may be stable across adulthood is synthesis capacity. Evidence from animals suggests that dopamine concentration declines with age even though synthesis may be preserved, so the increase in concentration into adulthood is also not quite right. Aside from synthesis, every other measure of DA changes from 18 to the end of life. See Karrer et al 2017 in Neurobiology of Aging. Figure 6 in that paper shows an aggregation of the subset of studies that have individual subject data and makes clear that the model of receptor plateau (depicted in the model diagram at the end of your paper) probably doesn't happen. Related to aging, in general the PET effects presented here are consistent with widely published adult age differences. For example, the lack of an age effect in ventral striatal D2 receptor availability has been documented recently in Seaman et al 2019 Human Brain Mapping.

We appreciate the reviewer bringing these highly relevant studies to our attention. It is very encouraging to see that our findings on receptor development, including the finding of diminished reductions in the NAcc are consistent with this recent work. These studies have now been incorporated to our Introduction and Discussion sections. Our schematic and description were largely based on rodent studies that focused on early development through early adulthood, and as such we had not introduced the important work in aging which clearly shows decreases in dopamine receptors into late life. We now remove language suggesting a “plateau” of receptor concentration in adulthood. We have also edited our schematic to 1) indicate we only depict development up to early adulthood, and 2) to reflect the non-linear, decelerating decrease into adulthood that is clearly described in Karrer et al 2017 and Seaman et al 2019 and reflects what we see in our own data. Our depiction of DA concentration was based on work such as (Giorgi et al., 1987; Rao et al., 1991) showing relative stability in DA concentration in rat striatum during early adulthood. Based on the comments of the reviewer we have also amended the schematized developmental curve for DA concentration to longer depict increases after early adulthood. Figure 3 now appears as follows:

Given the lack of a plateau, I was hoping for more discussion or even analysis of potential non-linear effects. In fact, there may be enough early adulthood (18-30 yo) data from the Karrer et al meta-analysis to examine nonlinearity during that phase - maybe for both D2 receptors and transporters. Those data are all publicly shared and open for secondary analysis.

We are certainly interested in non-linear effects, and we designed our statistical analysis pipeline to be sensitive to these non-linear age effects. Specifically, we perform a model selection procedure that compares linear, quadratic, and inverse age (age^{-1}) models which are sensitive to non-linear developmental effects. We now emphasize our rationale for this model selection procedure more explicitly in the method section:

“In order to detect potential non-linear effects of age, we first performed a model selection procedure in which we examined linear, quadratic, and age^{-1} functional forms of age and selected the best-fitting model using BIC. The best fitting functional form of age was used for subsequent modeling and significance testing.”

I didn't realize until the methods details that the rac scan was a task-displacement design. Individual differences in task ability can complicate the use of these measures as stable estimates of individual differences in receptor availability so it is good that the authors excluded the task part of the session. It's quite possible that 40 minutes of data approximates or maybe even almost exactly aligns with 60 minutes of data (which is common for raclopride). Do others have data showing this?

We are not aware of others showing the approximation between 40 and 60 minutes of raclopride data, though others have used as little as 15 minutes (Hamilton et al., 2018). We do now note however that this shorter window of time for data analysis may have undermined our ability to detect associations between R2' and raclopride in this dataset (e.g. this may have contributed to the wider confidence intervals for our raclopride – R2' coefficients). As such, we may have failed to detect a true effect, and future work is needed to replicate these associations.

Hamilton, J.P., ..., Gotlib, I.H., 2018. Striatal dopamine deficits predict reductions in striatal functional connectivity in major depression: a concurrent 11 C-raclopride positron emission tomography and functional magnetic resonance imaging investigation. *Translational Psychiatry* 8, 1–10.

Importantly, all participants completed the same raclopride acquisition protocol, so the number of minutes of acquired data should not systematically impact our developmental models. The fact that our observed developmental trajectories replicate earlier developmental work (including the papers the reviewer cited in the below comments), lends additional confidence.

I want to make it clear that this is an incredibly innovative study. There are multiple PET measures which is rare and the longitudinal data is incredibly valuable. I, likely much more than most, would be extremely excited if this association existed. I recently moved from an institution with a full PET center to one without local radiochem. I am desperate for MR measures that would provide reliable markers of dopamine function. I worry about the claims currently made in this paper though. The information presented does not seem strong enough to warrant the claims.

Disclaimer

Gregory Samanez-Larkin

We greatly appreciate the reviewer's enthusiasm for the innovative nature of the study and the value of the findings presented. As detailed above, we also appreciate the need for careful interpretation of the findings related to the association of R2' and DTBZ and have made substantial edits to the manuscript to more fully discuss the context, interpretation, and implications of these provocative results. We believe the changes implemented here have served to greatly strengthen our manuscript, and we thank the reviewer for his many thoughtful comments.

Reviewers' Comments:

Reviewer #1:

Remarks to the Author:

The authors have convincingly addressed the reviewers' points.

Reviewer #2:

Remarks to the Author:

The revised manuscript is much improved and the authors were highly responsive to feedback from all reviewers. The qualifications in the discussion are a step in the right direction and could be improved even more by being very clear about the uncertainty of the estimated association between R' and DTBZ. R' could be accounting for as much as 23% of the variance in DTBZ but as little as less than 1%. This is critically important to acknowledge; I'd consider adding it to the abstract. There are too many past examples of researchers basing countless subsequent claims off one initial moderate correlation in a small sample. My main worry is that no one else will do the PET. We'll get a proliferation of papers with dopamine in the title that only have R' when these data suggest that true effect could be very very small. Again, this is a highly novel paper that will have many interested readers. It's a very cool idea and an interesting set of findings.

Gregory Samanez-Larkin

REVIEWERS' COMMENTS:

Reviewer #1 (Remarks to the Author):

The authors have convincingly addressed the reviewers' points.

Reviewer #2 (Remarks to the Author):

The revised manuscript is much improved and the authors were highly responsive to feedback from all reviewers. The qualifications in the discussion are a step in the right direction and could be improved even more by being very clear about the uncertainty of the estimated association between R' and DTBZ. R' could be accounting for as much as 23% of the variance in DTBZ but as little as less than 1%. This is critically important to acknowledge; I'd consider adding it to the abstract. There are too many past examples of researchers basing countless subsequent claims off one initial moderate correlation in a small sample. My main worry is that no one else will do the PET. We'll get a proliferation of papers with dopamine in the title that only have R' when these data suggest that true effect could be very very small. Again, this is a highly novel paper that will have many interested readers. It's a very cool idea and an interesting set of findings.

Gregory Samanez-Larkin

We appreciate the kind comments from the reviewer and have incorporated his suggestion to add the effect size information to the abstract. The abstract now reads (added portion in **bold**):

We observed decreases in D2/D3 receptor availability with age, while presynaptic vesicular DA storage (DTBZ), which was significantly associated with R² (**standardized coefficient=0.29, 95% CI=[0.11, 0.48]**), was developmentally stable by age 18.